# The Chloroplast Genome of Endive (*Cichorium endivia* L.): Cultivar Structural Variants and Transcriptome Responses to Stress Due to Rain Extreme Events

**DOI:** 10.3390/genes14091829

**Published:** 2023-09-21

**Authors:** Giulio Testone, Michele Lamprillo, Maria Gonnella, Giuseppe Arnesi, Anatoly Petrovich Sobolev, Riccardo Aiese Cigliano, Donato Giannino

**Affiliations:** 1Institute for Biological Systems, National Research Council (CNR), Via Salaria Km 29,300, Monterotondo, 00015 Rome, Italy; giulio.testone@cnr.it (G.T.); michele.lamprillo@isb.cnr.it (M.L.); anatoly.sobolev@cnr.it (A.P.S.); 2Institute of Sciences of Food Production, National Research Council (CNR), Via G. Amendola 122/O, 70126 Bari, Italy; maria.gonnella@ispa.cnr.it; 3Enza Zaden Italia, Strada Statale Aurelia Km 96.400, Tarquinia, 01016 Viterbo, Italy; g.arnesi@enzazaden.it; 4Sequentia Biotech SL, C/ del Dr. Trueta 179, 08005 Barcelona, Spain; raiesecigliano@sequentiabiotech.com

**Keywords:** endive, cp genome sequence, DNA and RNA variants, cp transcriptome response, extreme rain stress

## Abstract

The chloroplast (cp) genome diversity has been used in phylogeny studies, breeding, and variety protection, and its expression has been shown to play a role in stress response. Smooth- and curly-leafed endives (*Cichorium endivia* var. *latifolium* and var. *crispum*) are of nutritional and economic importance and are the target of ever-changing breeding programmes. A reference cp genome sequence was assembled and annotated (cultivar ‘Confiance’), which was 152,809 base pairs long, organized into the angiosperm-typical quadripartite structure, harboring two inverted repeats separated by the large- and short- single copy regions. The annotation included 136 genes, 90 protein-coding genes, 38 transfer, and 8 ribosomal RNAs and the sequence generated a distinct phyletic group within *Asteraceae* with the well-separated *C. endivia* and *intybus* species. SSR variants within the reference genome were mostly of tri-nucleotide type, and the cytosine to uracil (C/U) RNA editing recurred. The cp genome was nearly fully transcribed, hence sequence polymorphism was investigated by RNA-Seq of seven cultivars, and the SNP number was higher in smooth- than curly-leafed ones. All cultivars maintained C/U changes in identical positions, suggesting that RNA editing patterns were conserved; most cultivars shared SNPs of moderate impact on protein changes in the *ndhD*, *ndhA*, and *psbF* genes, suggesting that their variability may have a potential role in adaptive response. The cp transcriptome expression was investigated in leaves of plants affected by pre-harvest rainfall and rainfall excess plus waterlogging events characterized by production loss, compared to those of a cycle not affected by extreme rainfall. Overall, the analyses evidenced stress- and cultivar-specific responses, and further revealed that genes of the Cytochrome b6/f, and PSI-PSII systems were commonly affected and likely to be among major targets of extreme rain-related stress.

## 1. Introduction

Endives are widely consumed in Europe as fresh or minimally processed and packaged salads with healthy properties [1]. They belong to the species *Cichorium endivia* (fam. *Asteraceae*) with the botanical varieties *crispum* and *latifolium*, which provide the genetic pools for the breeding of curly- and smooth-leafed commercial cultivars [2]. Endive is a low-input and cold-tolerant crop, although it is quite sensitive in terms of yield and quality to some stresses such as waterlogging or heavy rainfall [3]. More than 90% of the Italian production takes place outdoors (2037 tons in 2021; http://dati.istat.it (accessed on 13 September 2023)) and commercial varieties have been selected to guarantee quality standards under variable environmental conditions and continuous production cycles that overcome seasonality. In the current context of climate change, the intensification of unpredictable rainfall excess events (together with inadequate drainage systems and the soil’s intrinsic inability to store it) causes extreme damage to this crop even from short-term waterlogging [4]. Roots are the first anoxia-affected targets by soil anaerobiosis that trigger multi-level responses of the aerial organs resulting in stomatal closure, reduced leaf chlorophyll content and photosynthesis, and accelerated senescence [5]. The whole-plant response that alters nutrient contents is complex and varies with stress intensity and genotypes, for instance, a cultivar-specific carbohydrate drop occurred in endive challenged by natural short-term waterlogging [6]. Contextually, the chloroplast (cp) plays a central role in photosynthesis [7] and its genome responds to mechanical stresses/stimuli [8,9] such as water spray, precipitation hits and wounding of biotic or abiotic origin (e.g., pest bites or hail). Studies on the effects of heavy rainfall on plastid organelles are scarce and needed on a broad scale because of the expected increase in frequency and severity of extreme rainfall events [10]. Angiosperm cp genomes maintain a quadripartite structure [11,12]. However, the occurrence of sequence diversity [13] has allowed refined phylogenetic classification of the *Asteraceae* family into species, tribes, and subtribes [14,15]. Moreover, SNP markers have been useful for intraspecific genotyping with the aim of genetic breeding [12,16,17]. In this context, several resources (https://npgsweb.ars-grin.gov/gringlobal/search (accessed on 13 September 2023)) and markers are available for endive genetics [18], ranging from a reference genome sequence [19] and tissue-specific transcriptomes [6,20], although the cp genome sequence has not yet been published. Here, a chloroplast reference genome was assembled by DNA-Seq and structural variation in curly and smooth cultivars was further explored by RNA-Seq data. The cp transcriptome response was monitored in cultivars from two case studies of heavy pre-harvest rainfall (affecting production) compared to a standard cycle without extreme events. Gene expression analyses revealed a number of photosynthesis genes that showed similar expression changes, suggesting that they are common targets of extreme precipitation and may contribute to the resulting nutrient (sugar) loss.

## 2. Materials and Methods

### 2.1. Plant Material, Growth Conditions, Samplings and Studied Models

Enza Zaden Italy s.r.l. provided seeds and plants of curly- and smooth-leafed genotypes, which were patented cultivars and experimental lines, respectively identified by names and number codes. Curly-types: ‘Domari’ (D), ‘Myrna’ (M), ‘Imari’ (I), ‘A32861’ (A32); smooth-types: ‘Confiance’ (C), ‘Flester’ (F), ‘E02S.0338’ (E02S). The cultivation cycles spanned September-November 2011 and 2012; the dates of outdoor seedling transplant and harvest were 14/9/2011–17/11/2011 and 14/9/2012–15/11/2012. The growth occurred in the same parcels at Tarquinia (Lazio, Italy, 42°15’ N 11°44’ E, 31 m a.s.l.). Standard cultivation procedures, soil, water content variation, rainfall (R), relative humidity (RH), and temperature (T) were previously detailed [6,21,22]. In 2013, D and F only were grown in Conversano fields (41°00’ N 17°50’ E, 100 m a.s.l.) in soils with similar characteristics as Tarquinia’s [22] using standard interventions. Transplant and harvest dates were 15/9/2013–22/11/2013. Appendix A reports monthly average values of R, RH, and T of the entire cycle (September–November), while R daily values (Appendix A) and T weekly values (Appendix A) regard one month before harvest. The year 2011 was used as a reference because unexpected events of excess rain did not happen. Synoptically, R was much higher in 2012 than in 2011 (+200%) before harvest (November values in Appendix A), while a low increase of T and RH was recorded (+5 and +7%). The extreme rain event of 11/11/2012 (Appendix A) caused soil saturation (0 kPa) and root-zone waterlogging for 72 h, after which draining to field capacity (10 kPa) was restored [6]. In the cultivation of 2013, R was +49%, RH −3%, T +5% (November values in Appendix A), and the rainfalls of 12 and 19/11/2013 (Appendix A) did not cause waterlogging.

The transcriptomic analyses were conducted using D, M, C, and F grown in 2012, and D and F grown in 2013 compared to respective cultivars grown in 2011. In all experiments, sampling included 9 heads per cultivar, accurately selected according to market standards. The leaves (n = 10) intended for consumption were cut from each head and pooled into a cultivar-specific bulk, which was subdivided into 3 replicate batches (RB) of 30 leaves each. RBs of equal weights were frost in liquid nitrogen and gently crunched, and aliquots were ground to powder and used for nucleic acid isolation. As for variant studies based on RNA-Seq, all genotypes were used.

### 2.2. DNA Isolation, Sequencing, Chloroplast Assembly and Annotation

Total genomic DNA of ‘Confiance’ was isolated from two RBs using the DNeasy Plant Mini Kit (QIAGEN), checked for quality standards (A260/280 and A260/230 ratios > 1.8), and 1 µg was used for library construction and subjected to whole genome sequencing (IGA Technology Services, Udine, Italy). The Ovation^®^ Ultralow System V2 DNA-Seq Library Preparation Kit (Tecan, Männedorf, Switzerland) was used for library preparation according to the manufacturer’s instructions. Both input and final libraries were quantified using the Qubit 2.0 Fluorometer (Invitrogen, Carlsbad, CA, USA) and quality checked using the Agilent 2100 Bioanalyzer High Sensitivity DNA Assay (Agilent Technologies, Santa Clara, CA, USA). The library was sequenced in 150 bp paired-end mode on NovaSeq 6000 (Illumina, San Diego, CA, USA). Raw reads were checked for quality and filtered to remove ambiguous and poor-quality bases using FastQC v0.11.9 (www.bioinformatics.babraham.ac.uk/projects/fastqc/ (accessed on 13 September 2023)) and Trimmomatic v0.39, respectively. The chloroplast genome was assembled by NOVOPlasty v.4.2 [23] using the *C. intybus* chloroplast sequence (NCBI Reference Sequence: NC_043842.1) as reference genome, annotated by GeSeq [24] and visualised by OGDRAW [25], and cpgview [26]. The plastome sequence of *C. endivia* was stored in GenBank (Accession No. OQ928160). As for SSRs, MIcroSAtellite tool v1.0 (MISA; http://pgrc.ipk-gatersleben.de/misa (accessed on 13 September 2023)) scored sequence repeats (SSRs) targeting 1 to 10 nucleotide long stretches by minimum repetitions. The MISA script was set for SSR search as follows: “1-15 2-5 3-3 4-3 5-3 6-3” and “0” interruptions. SSRs were intersected with the endive annotation file (annotation.bed) using the options –wa and –wb.

### 2.3. RNA Isolation, Sequencing, SNP Calling and Transcriptomic Analyses

RNA was isolated from 3 replicate batch per genotype as described [21], yields and integrity (RIN > 7) were assessed (NanoDrop ND-1000, Thermo Scientific Inc (Waltham, MA, USA); BioAnalyzer 2100, Agilent Technologies Inc, cDNA libraries were synthesised from 1 µg of total RNA (TruSeq RNA-Seq kit, Illumina) and sequenced in 50 bp single-end mode (Illumina HiSeq2000; IGA Technology Services, Udine, Italy).

For variant identification, BWA (v0.7.17) was used to align the RNA reads to the reference genome, Picard tools (v 2.23.9) to remove PCR artefacts, GATK SplitNCigarReads (v4.2.6.1) to split reads containing Ns in their cigar string, and bcftools (v1.15.1) to identify raw variants. Biallelic SNPs were then recovered by filtering out variants with low genotype quality scores (GQ < 30), low alternative allele calling (QUAL < 20), inappropriate coverage depth (min: DP < 10; max: DP > 2*x*DP average), significant bias (strand, mapping quality, read position, and base quality). InDel consecutive clusters (distances < 10 bp) and SNPs close (±10 bp) to InDel were further excluded. Gene variant annotations and functional effect predictions were obtained using SnpEff (v5.1d).

For gene expression analyses, HISAT2 (ver 2.2.1, parameter “dta”) and StringTie (v2.2.1) were respectively used to map the reads on the reference genome and assemble them into transcripts. Transcript-level count data were used to estimate gene-level abundances by the R-package Tximport (v.1.28.0). Analysis of differentially expressed genes (DEG) was achieved by DESeq2 package [27] and by selecting genes with false discovery rate (FDR) ≤ 0.05 and an absolute log2 fold change ≥1.

### 2.4. Phylogenetic Trees

A phylogenetic tree was constructed using 17 chloroplast genome sequences from species (listed below) of the *Cichorieae* tribe. MAFFT v.7.490 (https://mafft.cbrc.jp/alignment/software/ (accessed on 13 September 2023)) produced sequence alignment (Appendix A) and MEGA11 (https://www.megasoftware.net/ (accessed on 13 September 2023)) produced both the distance matrix and the Maximum Likelihood tree with 1000 bootstrap iterations. The phylogenetic tree of botanical varieties and cultivars of endive was built using SNP concatemers, which generated FASTA sequence files further processed by MEGA11. *Cicerbita alpina* Wallr. (NC_066724.1), *Cichorium endivia* L. (OQ928160), *Cichorium intybus* L. (NC_043842.1), *Crepidiastrum sonchifolium* (Maxim.) Pak and Kawano (NC_046513.1), *Crepis japonica* (L.) Benth. (NC_046516.1), *Ixeris polycephala* Cass. (NC_046514.1), *Lactuca sativa* L. (NC_007578.1), *Lapsanastrum humile* (Thunb.) Pak and K.Bremer (NC_046515.1), *Launaea arborescens* (Batt.) Murb. (NC_060827.1), *Notoseris macilenta* (Vaniot and H.Le’v.) N.Kilian (NC_066766.1), *Paraprenanthes melanantha* Ze H.Wang (NC_066769.1), *Reichardia picroides* (L.) Roth (NC_062408.1), *Sonchus arvensis* L. (NC_054161.1), *Sonchus brassicifolius* S.-C.Kim and Mejias (NC_051922.1), *Soroseris hookeriana* Stebbins (NC_070135.1), *Stebbinsia umbrella* (Franch.) Lipsch. (NC_051973.1), *Taraxacum officinale* F.H.Wigg (NC_030772.1), and *Cucumis sativus* L. (NC_007144.1) as outgroup.

### 2.5. Metabolite Profiling of Hydro-Soluble Compounds by NMR

The whole methodology to achieve water-soluble fraction starting from 25 mg of lyophilized ground leaves of endives and producing both NMR spectra (Bruker AVANCE 600 NMR; proton freq. of 600.13 MH) and metabolite concentrations were detailed previously [6].

## 3. Results

### 3.1. Endive Chloroplast DNA Assembly and Annotation

After achieving 161,817,528 clean reads (95.4%) counting for more than 24.3 billion bases from high-throughput sequencing (Q20 and Q30 were 99.99% and 97.3%, respectively), a chloroplast genome of *C. endivia* was assembled showing a length of 152,809 bp and an organization into a typical quadripartite structure. It comprised two inverted (IRA and IRB), one large single-copy (LSC), and one short single-copy (SSC) region (Figure 1), which, respectively, consisted of 25,085, 84,057 and 18,582 bp, with an overall GC content of 37.7% (Table 1). The genome encoded 136 genes, consisting of 90 protein-coding genes (8 lay in both IRs), 38 transfer RNAs (tRNAs; 8 in both IRs), and 8 ribosomal RNAs (rRNAs; 4 in both IRs). The set of 116 unique genes (79 protein-coding, 30 tRNAs, 4 rRNAs, and 3 pseudogenes) was annotated and grouped according to shared functions. Overall, 16 unique genes bore one (*atpF, ndhA, ndhB, petB, petD, rpl16, rpl2, rpoC1, rps12, rps16, trnA-UGC, trnE-UUC, trnG-UCC, trnK-UUU, trnL-UAA, trnV-UAC*), or two introns (*pafI* and *clpP*, Appendix A). Among intron-containing genes, 12 genes (*atpF, clpP, pafI, petB, petD, rpl16, rpoC1, rps16, trnG-UCC, trnK-UUU, trnL-UAA, trnV-UAC*) were in the LSC, one gene (*ndhA*) fell in the SSC, 5 genes (*rps12, ndhB, rpl2, trnA-UGC*(x2), *trnI-GAU*) occurred in both IRs, and the trans-splicing *rps12* gene had the exon 1 in the LSC while the exon 2 and 3 in the IR regions (Appendix A). As for the region boundaries, the *rps19* and ^Ψ^*ycf1* respectively hosted the LSC/IRB and IRB/SSC junctions; the *ycf1* and ^Ψ^*rps19* the SSC/IRA and IRA/LSC junctions. The trnN–GUU/*ndhF* and *rpl2*/trnH–GUG intergenic regions respectively harboured the IRB/SSC and IRA/LSC junctions (frequently found feature).

### 3.2. Phylogenetic and Sequence Variant Analyses

The ‘Confiance’ chloroplast genome sequence was included in a phylogenetic tree construction that was restricted to species of *Cichorioideae* subfamily (Figure 2) to address whether chloroplast DNA polymorphisms could separate the subtribe groups. The genus *Cichorium* formed a phyletic group per se with the well-separated *endivia* and *intybus* species at the level of subtribes. Structurally, the ‘Confiance’ cp genome harboured SSR variants (Table 2 and Table 3) and RNA editing sites (Appendix A). Di-, tri-, and tetra-nucleotides were scored a maximum of six times (Table 2); briefly, the three-time repeated trinucleotides were the most frequent (65 times) and the AAT/ATT motifs prevailed (34.3%). Moreover, out of 48 SSRs, 43 fell in 24 in mRNAs, 3 in tRNAs, and 2 in rRNAs, and the *ycf2* genes contained the highest SSR number, followed by the *ndh* group (Table 3).

Cytosine to uracil (C/U) changes are typical of *Asteraceae,* as variant calling on the DNA-Seq data of ‘Confiance’ did not reveal any variants (not shown), DNA/RNA-Seq comparison within the cultivar (Appendix A) identified 20 C/U events (homo + het) out of 104 SNPs (Table 4) falling in 12 protein-coding genes, and all cultivars showed C/U homozygous switches in 13 identical positions (Appendix A, grey shaded rows), suggesting that RNA editing was conserved.

Nucleotide variation was further explored using RNA-Seq data of seven cultivars that showed 70 to 98% coverage vs. the ‘Confiance’ reference genome (Figure 3 and Table 4). Focusing on cultivars with >90% coverage, the variant number was higher in smooth (C and F) than curly (M and D) ones, 141–152 vs. 113–129. As for all cultivars, the normalised frequency of variants (total variants/RNA-Seq covered region) ranged from 0.79 in M to 1.08 in A32 (Table 4). A phylogenetic tree was built by SNP concatemers using 7 cultivars; separation into the curly and smooth botanical varieties was observed though supported by modest bootstrap values (Appendix A).

As for the SNP putative effects (Table 5), 10 were intergenic and 41 lay in coding regions; variants with high and moderate impacts were ca. 55%, the high ones (5.9%) were of stop-lost/gained types, whilst the moderate ones (49%) were all missense. Private SNPs were assigned (Figure 4) in those regions that were fully covered by the reads of all cultivars and found as heterozygous type only in I (2), M (1), and E02S (5). The private SNPs of M, I, and E02S (1, 2, and 4, respectively) fell into protein-coding genes, and a high-impact variant occurred in *rpl14* of Imari (Table 6). Among all cultivars, six genes (Table 7) contained homozygous SNPs of high or moderate impact that are predicted to cause protein changes; specifically, the *ndhD*, *ndhA*, *psbF* genes shared moderate events in 6 out of 7 cultivars.

### 3.3. Cp Transcriptome Response to Heavy Rain-Related Stress Before Harvest

Two case studies related to cultivation that underwent rain excess before harvest (Appendix A) were used to compare leaf cp expression with that of a cycle unaffected by extreme events. As for the high vs. moderate rainy year (HRY vs. MRY), a 60% rain surplus occurred in November (Appendix A, bold values), and loss of several nutrients was recorded in D and M cultivars by NMR survey (Appendix A), with significant impact on carbohydrate levels. In the second case study, cultivars D, M, C, and F were affected by excessive rainfall and waterlogging as described above [6].

Looking at the cp transcriptome variation of HRY vs. MRY (Table 8, left columns), we scored 14 differentially expressed genes (DEGs) with modest fold change variation (min and max values were −1.27 and 1.51). The smooth F was more responsive than the curly D (13 vs. 8 DEGs); both cultivars shared the up-regulation of 2 genes belonging to cytochrome b_6_f (Cyt.b6/f) and 2 of the large subunit ribosomal protein groups (LS-RP), and the down-regulation of 3 genes of the photosystem I and II groups (PSI and II). Moreover, it was observed the cultivar F-specific variation of 6 genes, which were scattered in the groups of NADH-oxidoreductases (NADH-OR), PSI and II, RNA polymerases (RNA-POL), and small subunits ribosomal protein (SS-RP), while cultivar D-specific variation included 1 gene of the PSI group.

As for the cp expression in HRY + WL vs. MRY (Table 8, right columns), a total of 25 genes among the four cultivars showed expression variation ranging from −2.14 to 2.08 fold changes; the smooth C and F were more responsive than curly D and M (18–22 vs. 13–14 DEGs). All cultivars shared the down-regulation of 5 genes within the PSI and PSII groups, and the up-regulation of 3 genes of the Cyt.b6/f and 1 (very modest) of the NADH-OR group; cultivar-specific expression also occurred (e.g., *rpoC2* and *rpl18* were unique DEG in ‘Confiance’). Overall, the altered expression pattern recurred in genes of the Cyt.b6/f, PSI, and PSII systems in the two case studies, suggesting that these may be among the major targets of extreme rain.

## 4. Discussion

The endive chloroplast genome here sequenced is conformed in the typical quadripartite structure [11,12] and its size falls in the range of species of the *Cichorieae* tribe [28]. Narrowing comparisons between the *C. intybus* and *endivia* close species, we observed that the latter showed a 166 bp smaller genome [29] due to shorter LSC (−319 bp) and longer IR and SSC (21 and 132 bp), higher n. of genes (136 vs. 127), of protein coding ones (90 vs. 74), and of tRNAs (38 vs. 29), and lower n. of rRNAs (8 vs. 24). Moreover, *pafI* and *clpP* contained two introns in both species, which also showed similar GC content (37.7 vs. 37.3%). Chloroplast genome variation within an individual (heteroplasmy) can be due to variable orientation of SSC and IR zones (defining distinct haplotypes) or due to DNA sequence polymorphism [30,31]. The endive plastome was constructed by short reads and did not allow the assembly of two haplotypes, which are known to occur in lettuce through long-read sequence technology [30]. However, several polymorphic events were revealed within the same cultivar by multiple RNA seq runs. Hence, the SNP pools that existed after selecting out the RNA editing motifs known for *Asteraceae* [32] indirectly support the sequence heteroplasmy as observed in cultivars of other crops [33,34].

In most species of *Asteraceae*, the positions of *rps19* and ^Ψ^*rps19* respectively mark the LSC/IRB and IRA/LSC junctions and those of ycf1 and ^Ψ^*ycf1* mark the IRB/SSC and SSC/IRA junctions [28]. Here (Figure 1 and Figure 3), the endive plastome sequence showed swapped positions for *ycf1* and ^Ψ^*ycf1*, as observed both in *Asteraceae* [35] and other species [36], indicating that the SSC orientation may belong to the haplotype A [30]. After refining the annotation of both *rps19* and ^Ψ^*rps19* from *C. intybus* (NC_043842.1), we observed that both species shared the same orientations. In coding regions, *ycf2* showed a high SSR number in both IRs, consistently with other *Asteraceae* species [35,37] representing a region of variability useful to develop highly specific markers. Cp sequence variability allowed the separation of several species at the subtribe level so that *Cichoriinae* fell apart from *Lactucinae*, *Crepidinae,* and *Hyoseridinae*, which is consistent with the effective use of chloroplast sequence to place subtribes within genera [38]. The 98.4% RNA-Seq coverage vs. cp genome in Confiance is consistent with the almost full transcription in plants [39,40] and that of the other cultivars ranged from 74 to 96%, scoring variants inclusive of private SNPs. The association between gene variants and cultivar leaf-type can be attributed to endive breeding programs based on crossing the same types rather than a functional impact on leaf shape. The trend of SNP number was higher in smooth- than curly- types though a much larger variety pool is needed to confirm it. Most cultivars commonly hosted SNPs of moderate effect (missense) in *ndhA, D,* and *psbF* genes. The *ndhA* and *psbF* of some species undergo RNA editing and transcription slippage that explain variability [41]; Table 6 of SNP impact excluded C/U editing, hence, the computed protein isoforms may account for diversified functions related to stress response/protection as shown for *ndh* genes [42]. The *ycf1* hosts frequent nonsynonymous SNPs, which were observed in endives and are often used in barcoding alternatively to the *matK* and *rbcL* genes [43]. Similarly, the *rpo* genes are relatively fast-evolving sequences used for marker production [44]; however, SNPs in endive *rpoC1* suggest a high functional impact that prompts further investigation.

Here, the affected leaf nutrient content confirmed that endive is a sensitive crop to excessive rainfall before harvest [6]; therefore, it is useful to gain insight into the cp transcriptome response of cultivars under this multifactorial stress. In both case studies, the smooth types showed a higher number of responding genes than the curly ones, and the range of expression variation was higher in all cultivars that underwent waterlogging. The former results suggest the occurrence of cultivar-dependent plasticity, which has also been observed in stressed lettuce [45] and deserves further study. The latter results may be due to the severity and/or complexity of the stress. The similar expression patterns observed in the two stress situations (one of which was exacerbated by waterlogging) could be the result of different transduction and control processes, considering the mutual interactions between chloroplast and nucleus [46]. Consistently, waterlogging is known to affect chloroplast morphology in sensitive lines [47] accompanied by the involvement of gene expression re-arrangements. In this work, the up-regulation of *RPL20* and *RPL33* in D and M cultivars is consistent with the stress status of the chloroplast; the former gene is essential for ribosome assembly and the latter is highly sensitive to temperature stress [48]. Moreover, the upregulation of Cyt.b6/f genes is accompanied by the down-regulation of some PSI and PSII ones in both case studies, supporting that these are common targets of the two stresses. Cytochrome b6/f controls the electron transfer from PSII to PSI and alleviates oxidative stress [49], so the altered and “orchestrated” expression would sustain that the cp machinery was undergoing photosynthetic impairment and oxidative stress. Consequently, it is proposed that such an alteration contributes to the loss of leaf carbohydrate content, although further studies targeting gene function are needed.

## Figures and Tables

**Figure 1 genes-14-01829-f001:**
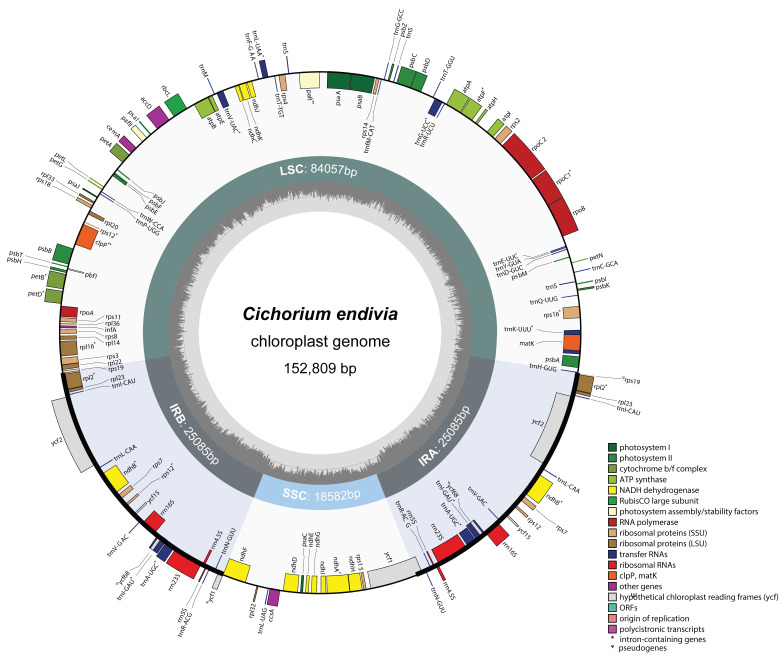
Circular chloroplast genome map of C. endive cv ‘Confiance’. GC contents are graphed in the inner circle reports. The large single-copy (LSC), the short single-copy (SSC), and the two inverted regions with their relative sizes are marked in the middle circle. Genes lie in the outer circle and are colored according to different colour-coded functional groups. Genes drawn outside or inside the circle are transcribed clockwise or anticlockwise, respectively.

**Figure 2 genes-14-01829-f002:**
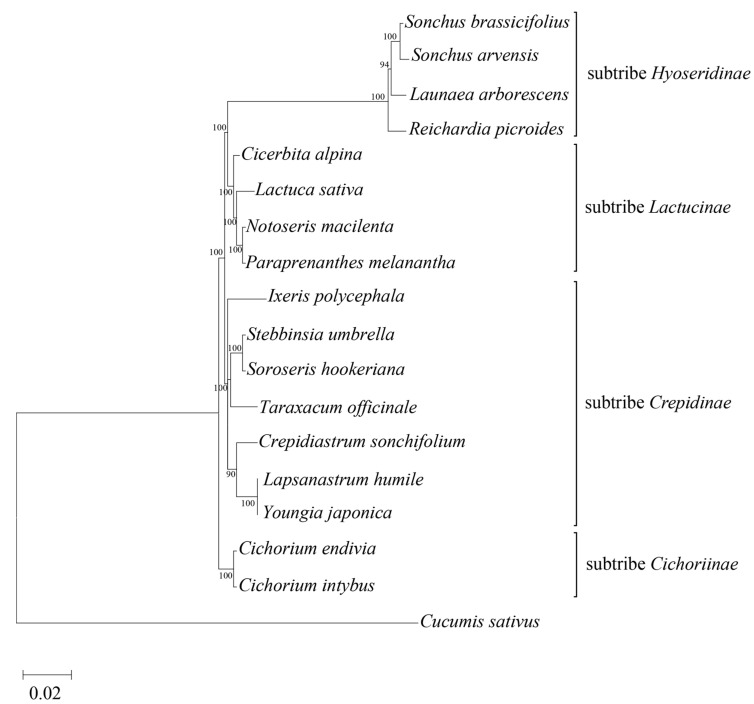
Phylogenetic analysis. The maximum likelihood tree of species belonging to the *Cichorieae* tribe was built using their complete chloroplast genomes (accession numbers are in the material and methods section). The chloroplast sequence of *Cucumis sativus* (fam. *Cucurbitaceae*) was used as an outgroup. The bootstrap values were inferred from 1000 replicates. The scale bar reports the number of changes per site.

**Figure 3 genes-14-01829-f003:**
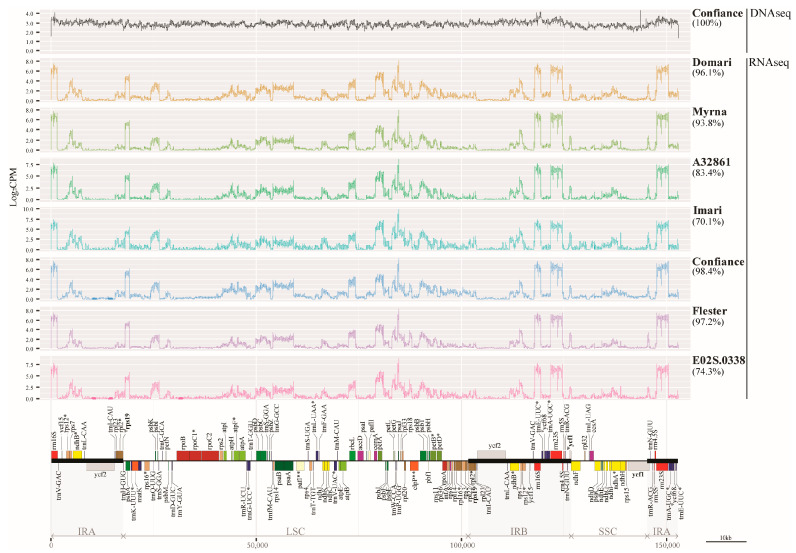
Coverage plots. Top: coverage of DNA-Seq (black line) and RNA-Seq (coloured lines) data over the reference cp genome measured as the logarithm of counts per million reads aligned over a given region. Bottom: linearised map of the cp genome. Genes marking region boundaries are bolded. *, Single intron gene; **, gene containing two introns; Ψ, pseudogene.

**Figure 4 genes-14-01829-f004:**
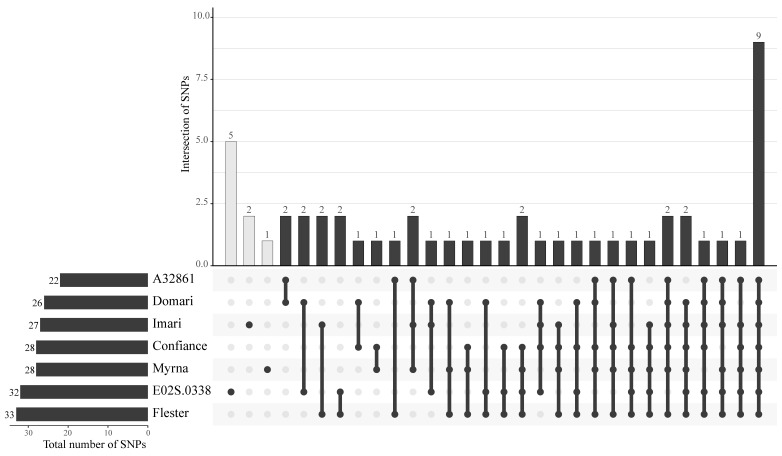
UpSet plot of private and shared SNPs among the seven cultivars. In the upper panel, the light and dark grey vertical bars correspond to the number of SNPs that are genotype-specific or shared by two or more genotypes, respectively. The number of SNPs in each set is given above the bars. Single and connected dots indicate the cultivar (names are on the left panel) that shares each gene group. Horizontal bars indicate the total number of SNP per cultivar.

**Table 1 genes-14-01829-t001:** Summary of the complete genome features of *C. endivia* chloroplast.

Parameters	Values	Gene Groups
Total length (bp)	152,809	
LSC size (bp)	84,057	
SSC size (bp)	18,582	
IR size (bp)	25,085	
GC%	37.7%	
Genes number	136 (20 occurring in both IRs)	
Protein-coding genes	90 (8 duplicated in IR)	
	*psaA, psaB, psaC, psaI, psaJ, pafI^**^(ycf3), pafII(ycf4),*	Photosystem I
	*psbA, psbB, psbC, psbD, psbE, psbF, psbH, psbI, psbJ, psbK, psbM, pbf1, psbT, psbZ*	Photosystem II
	*petA, petB *, petD *, petG, petL, petN*	Cytochrome b6/f complex
	*ccsA*	Cytochrome C synthesis
	*atpA, atpB, atpE, atpF *, atpH, atpI*	ATP synthase
	*rbcL*	RuBisCO
	*ndhA *, ndhB ** (x2)*, ndhC, ndhD, ndhE, ndhF, ndhG, ndhH, ndhI, ndhJ, ndhK*	NADH oxidoreductase
	*rpl2 ** (x2)*, rpl14, rpl16 *, rpl20, rpl22, rpl23* (x2)*, rpl32, rpl33, rpl36*	Large subunit ribosomal proteins
	*rps2, rps3, rps4, rps7* (x2)*, rps8, rps11, rps12 *^,a^* (x2)*, rps14, rps15, rps16 *, rps18,* ^Ψ^*rps19, rps19*	Small subunit ribosomal proteins
	*rpoA, rpoB, rpoC1 *, rpoC2*	RNA polymerase
	*infA*	Translation initiation factor
	*accD, cemA, clpP **, matK*	Others
	^Ψ^*ycf1, ycf1, ycf2* (x2)*, ycf15* (x2), ^Ψ^*ycf68 (x2)*	Unknown function genes
tRNA genes	38 (8 occurring in both IRs)	
	*trnA-UGC ** (x2)*, trnC-GCA, trnD-GUC, trnE-UUC* (x2)*, trnF-GAA, trnfM-CAT, trnG-GCC, trnG-UCC ^*^, trnH-GUG, trnI-GAU * (x2), trnI-CAU* (x2)*, trnK-UUU *, trnL-CAA* (x2)*, trnL-UAA *, trnL-UAG, trnM-CAU, trnN-GUU* (x2)*, trnP-UGG, trnQ-UUG, trnR-ACG* (x2)*, trnR-UCU, trnS-GCU, trnS-GGA, trnS-UGA, trnT-GGU, trnT-UGU, trnV-GAC* (x2)*, trnV-UAC *, trnW-CCA, trnY-GUA*	Transfer RNAs
rRNA genes	8 (4 occurring in both IRs)	
	*rrn4.5S* (x2)*, rrn5* (x2)*, rrn16* (x2)*, rrn23* (x2)	Ribosomal RNAs

*, Single intron gene; **, Gene containing two introns; (x2), indicates genes duplicated in the IR regions. a, Trans-splicing gene. Ψ*,* pseudogene.

**Table 2 genes-14-01829-t002:** SSR in cp of Confiance.

Unit Repeat Type	Number of Repetitions				Total	Major Type (%)
	3	4	5	6		
Di-nucleotide	0	0	1	1	2	AT/AT (100.0%)
Tri-nucleotide	65	2	0	0	67	AAT/ATT (34.3%)
Tetra-nucleotide	1	0	0	0	1	AAAG/CTTT (100.0%)

**Table 3 genes-14-01829-t003:** Distribution of SSR in cp genes.

Type	Gene	#SSR	Type	Gene	#SSR
mRNA	*ycf1* (IRA)	2	mRNA	*rps19* (IRB)	1
	*ycf2* (IRA)	5		*ndhA*	3
	*ycf2* (IRB)	5		*ndhB* (IRA)	2
	*accD*	1		*ndhB* (IRB)	2
	*matK*	2		*ndhF*	2
	*rpoC1*	2		*psbA*	1
	*rpoC2*	1		*psbB*	2
	*atpI*	1		*psbC*	2
	*rbcL*	1		*psaA*	1
	*rpoA*	2		*psaB*	1
	*rpl16*	1	tRNA	*trnK-UUU*	2
	*rpl22*	1		*trnV-UAC*	1
	*rpl36*	1	rRNA	*rrn23S* (IRB)	1
	*rps18*	1		*rrn23S* (IRA)	1

**Table 4 genes-14-01829-t004:** SNP overview.

	Smooth			Curly			
Event	Confiance	Flester	E02S	Domari	Myrna	Imari	A32861
No calls	18	12	43	22	46	59	10
HomoREF	41	36	40	49	41	39	52
Het	66	72	53	59	45	40	63
HomoALT	75	80	64	70	68	62	75
**TOT variants** ^a^	**141**	**152**	**117**	**129**	**113**	**102**	**138**
SNP	104	108	87	95	83	79	98
INDEL	37	44	30	34	30	23	40
Coverage (%)	98.4	97.2	74.3	96.1	93.8	70.1	83.4
Coverage (kbp) ^b^	150.36	148.53	113.54	146.85	143.33	107.12	127.44
Freq (variants/kb)	0.94	1.02	1.03	0.88	0.79	0.95	1.08
Freq (bp/variants)	1066.4	977.2	970.4	1138.4	1268.4	1050.2	923.5

a, Total variants account for the sum of heterozygous and homozygous for alternative allele genotypes. b, Normalised with respect to the length of the Confiance cp genome.

**Table 5 genes-14-01829-t005:** Overview of SNP effects.

Impact	n.	%	Effect	n.	%
High	3	5.9	stop_lost	2	3.9
			stop_gained	1	2.0
Moderate	25	49.0	missense_variant	25	49.0
Low	5	9.8	synonymous_variant	4	7.8
			splice_region_variant&intron_variant	1	2.0
Modifier	18	35.3	intergenic_region	10	19.6
			intron_variant	7	13.7
			non_coding_transcript_exon_variant	1	2.0

**Table 6 genes-14-01829-t006:** Private SNPs in genes.

Gene	Pos	Ref	Alt	Impact	Effect	Myrna	E02S	Imari
*psbC*	51406	C	A	Moderate	Missense_variant		*	
*psaB*	55447	A	G	Moderate	Missense_variant			*
*cemA*	78984	A	G	Moderate	Missense_variant		*	
*psbE*	82115	G	C	Moderate	Missense_variant		*	
*psbE*	82130	G	A	Moderate	Missense_variant		*	
*rpl14*	98501	C	A	High	stop_gained			*
*ndhA*	137326	C	T	Modifier	intron_variant	*		

*, SNP occurrence.

**Table 7 genes-14-01829-t007:** Homozygous SNPs and likely effect on deduced proteins.

	Cultivars ^a^							SNP Effect ^b^	SNP Impact ^c^	Protein Changes ^d^
Genes	C	F	E	D	M	I	A			
*rpoC1*	*			*		*	*	stop-	H	Ter598Gluext *?
*psbZ*						*		miss	M	Ser17Leu
*psbF*	*	*	*		*	*	*	miss	M	Ser26Phe
*ndhD*	*	*	*	*	*	*		miss	M	Ser160Leu
*ndhA*	*	*	*	*	*	*		miss	M	Ser321Phe
*ycf1*	*	*	*				*	miss	M	Phe664Leu

a, Smooth-leafed: C, Confiance; F, Flester; E, E02S.0338. Curly-leafed: D, Domari; M, Myrna; I, Imari; A, A32861. b, SNP effects: stop-, stop-lost; Miss, missense variant. c, SNP impacts. H, high; M, moderate. d, Sequence variant annotation in the format: “reference amino acid” “position” and “changed amino acid”. *, SNP occurrence. The notation “Ter598Gluext*?” refers to the loss of the normal termination site, occurring at position 598, its substitution by a Glu-codon, and the addition of a tail of new amino acids of unknown length (position *?), since the shifted frame does not contain a new stop codon.

**Table 8 genes-14-01829-t008:** Cp gene expression variation in years affected by heavy rain.

Gene	Group	HRY vs. MRY				HRY + WL vs. MRY							
		D		F		D		M		C		F	
		LFC	Padj	LFC	Padj	LFC	Padj	LFC	Padj	LFC	Padj	LFC	Padj
*atpI*	ATP-SYN									−1.01	*	−1.19	**
*atpE*										−1.18	*	−1.05	*
*atpB*						−1.28	**					−1.04	*
*petA*	Cyt.b6/f	**1.01**	******	**1.07**	******	**1.36**	*******	**1.02**	*****	**1.40**	*******	**1.55**	*******
*petL*						**1.35**	******	**1.10**	*****	**1.52**	******	**1.52**	*******
*petG*		**1.04**	******	**1.07**	*****	**1.83**	*******	**1.50**	******	**1.70**	******	**1.83**	*******
*rpl33*	LS-RP	**1.16**	*****	**1.04**	******	**1.07**	*****			**1.16**	******	**1.41**	*******
*rpl20*		**1.40**	*****	**1.36**	*******	**1.33**	******			**1.30**	******	**1.12**	******
*ndhB*	NADH-OR			**1.40**	******			**1.08**	*****	**1.09**	*****	**1.28**	*******
*ndhJ*						1.08	**	1.02	**	1.08	**	1.04	**
*ndhB*				**1.34**	******					**1.10**	*****	**1.23**	*******
*ndhH*				−1.12	*								
*psaB*	PSI	**−1.09**	******	**−1.03**	******	**−1.16**	*******	**−1.07**	*****	**−1.08**	*****	**−1.39**	*******
*psaA*		**−1.13**	******	**−1.23**	*******	**−1.08**	******	**−1.01**	******	**−1.46**	*******	**−1.52**	*******
*psaJ*		**1.02**	*******			**2.08**	*******			**1.63**	*******		
*psbA*	PSII	**−1.13**	******	**−1.27**	******	**−1.04**	*****	**−1.58**	*******	**−1.04**	*****	**−2.14**	*******
*psbI*				**1.19**	******			**1.00**	*****			**1.82**	*******
*psbD*						−1.16	**	−1.00	*	−1.08	*	−1.20	***
*psbC*						−1.05	**	−1.04	*	−1.11	*	−1.12	***
*psbZ*								−1.45	**			−1.37	***
*psbE*								−1.10	*			−1.16	*
*rpoC2*	RNA-POL			**−1.25**	******							**−1.10**	*****
*rbcL*	RuBisCO							−1.18	*			−1.09	*
*rps7*	SS-RP									1.12	**		
*rps16*										1.26	*		
*rps18*				**1.51**	*******							**1.47**	*******

Log fold change (LFC) values are negative and positive respectively in down- and up-regulated genes vs. control (year with no unexpected heavy rain events). Expression trends shared among all cultivars within heavy rain (HRY) or waterlogging events (HRY+WL) are shaded in grey. Gene expression trends conserved in both events are bolded. ***, **, * respectively significant for *p* < 0.01, 0.05, 0.1. ATP-SYN, ATP synthases; Cyt.b6/f, Cytochrome b6/f complex; LS-RP, Large subunit ribosomal proteins; PSI, Photosystem I; PSII, Photosystem II; RNA-POL, RNA polymerase; SS-RP, Small subunit ribosomal proteins.

## Data Availability

The chloroplast genome sequence of *Cichorium endivia* has GenBank accession OQ928160.

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
