# Peer review of "The Chloroplast Genome of Endive (Cichorium endivia L.): Cultivar Structural Variants and Transcriptome Responses to Stress Due to Rain Extreme Events"

_genes, 2023, doi:10.3390/genes14091829_

Round 1

Reviewer 1 Report

I checked your manuscript and described comments below.

The diversity of chloroplast genomes is an extremely important factor for cultivating plants. This paper performs a very good transcriptome analysis of genes expressed in chloroplasts.

I have a question regarding the following points.

1.       Phylogenetic analysis uses MEGA X. Currently it is MEGA11. It's better to use new software.

2.       The number of bootstrap values ​​in Figure 2 is not written in the method.

3.       It is better to include the alignment results by Mafft in the Supplementary file if possible.

I don't think this paper has any major mistakes or grammatical problems.

Author Response

We thank the Reviewer 1 for useful suggestions and comments.

Here below point-by-point replies.

1. Phylogenetic analysis uses MEGA X. Currently it is MEGA11. It's better to use new software.

MEGA11 was used to construct new phylogenetic trees which did not substantially differ from the previous ones.

2. The number of bootstrap values in Figure 2 is not written in the method.

We added the following sentence accordingly:

New lines 153-154 (Material and methods): “…the Maximum Likelihood tree with 1000 bootstrap iterations.”

New lines 260-261 (figure 2 legend): “The bootstrap values were inferred from 1000 replicates.”

3. It is better to include the alignment results by Mafft in the Supplementary file if possible.

Alignment results by MAFFT were added in the Data S1

Reviewer 2 Report

One more polishing of English is necessary. A few questions to improve the discussion. Why do authors need phylogenetic analysis? It cannot be exact due to the shallow sampling. When citing the first time, all taxonomic names must be accommodated by the authors' names. Why cultivars, not cryptic taxonomic entities? Variants in sequences must be treated more seriously. It is logically unclear in what way sequence variation of the plastomes can be the response to something. The name "endive" must be joined with the proper binomial right in the paper's title.

Minor editing of the English is required.

Author Response

We thank the Reviewer 2 for fully recommending the work and providing good points to ameliorate the discussion. Here below point-by-point replies.

1. One more polishing of English is necessary.

We found your comments and suggestions useful to revise not only the discussion but also the results regarding Figure 2B, which was presented as a supplementary figure (Figure S3), and the texts of the Abstract, Results and Discussion were slightly modified accordingly, further polishing the English text.

2. Why do authors need phylogenetic analysis? It cannot be exact due to the shallow sampling.

The phylogenetic tree focused on the tribes of the Cichorioideae subfamily (as specified in new text, see below) to test whether chloroplast DNA polymorphisms could separate the subtribe groups. Given the large number of available accessions whose cpDNA was sequenced, we randomly selected one of high quality per each species to resolve the classification at the subtribe level, which was successful within this sample size.

Lines 191-192 (Results). The previous sentence “The 'Confiance' chloroplast genome sequence was included in a phylogenetic tree construction that embraced genera of Cichorioideae subfamily (Figure 2A)”

was replaced with

“The 'Confiance' chloroplast genome sequence was included in a phylogenetic tree construction that was restricted to species of Cichorioideae subfamily (Figure 2) to address whether chloroplast DNA polymorphisms could separate the subtribe groups.”

3. When citing the first time, all taxonomic names must be accommodated by the authors' names.

Authors' names of species used in phylogenetic analysis were added in Material and methods, par. 2.4.

4. Why cultivars, not cryptic taxonomic entities?

We interpret the sentence as follows: “Why did you use cultivars for the phylogenetic tree? Are not these cryptic taxonomic entities?”. We believe that cultivars (cultivated and patented varieties) are difficult to distinguish at the phenotypic level. Therefore, the cultivar-based phylogenetic tree was previously built to ascertain whether chloroplast DNA variants were effective in grouping cultivars into their respective botanical varieties. We further realised that bootstrap values of the phylogenetic tree (based on SNP concatemers) was feeble and decided to present it as supplementary material (Figure S3). Subsequently, the Abstract, Results and Discussion were edited accordingly.

Line 24 (Abstract). The text “, and SNP concatemers were effective to phylogenetically separate the two morpho-types” present in the previous version was deleted.

Lines 238-239 (Results). The sentence “Subsequently, a phylogenetic tree was built by SNP concatemers, that were able to split the 7 cultivars into the curly and smooth groups (Figure 2B).”

was replaced by:

“A phylogenetic tree was built by SNP concatemers using 7 cultivars; separation into the curly and smooth botanical varieties was observed though supported by modest bootstrap values (Figure S3).

 Lines 367-370 (Discussion). The sentence “The 98.4% RNA-Seq coverage vs cp genome in Confiance is consistent with the almost full-transcription in plants [37,38] and that of the other cultivars ranged from 74 to 96%, scoring SNP variants effective in separating smooth from curly types and in scoring private SNPs”

was modified as:

“The 98.4% RNA-Seq coverage vs cp genome in Confiance is consistent with the almost full-transcription in plants [39,40] and that of the other cultivars ranged from 74 to 96%, scoring variants inclusive of private SNPs.”

5. Variants in sequences must be treated more seriously. It is logically unclear in what way sequence variation of the plastomes can be the response to something.

We are afraid that we do not know where we reported that "sequence variation of the plastomes could be the answer to something". The manuscript deals with the concept that allelic variants causing protein (and functional) variation may contribute to diversified adaptation responses to stress. Identifying these variants is important for molecular breeding.

In order to better clarify this concept, the text in lines 26-27 “most cultivars shared SNPs of moderate impact on protein changes in the ndhD, ndhA, psbF genes, suggesting that these are targets with a potential role in adaptive response”

was improved in:

“most cultivars shared SNPs of moderate impact on protein changes in the ndhD, ndhA, psbF genes, suggesting that their variability may have a potential role in adaptive response”

6. The name "endive" must be joined with the proper binomial right in the paper's title.

The title was changed as follows:

“The chloroplast genome of endive (Cichorium endivia L.): cultivar structural variants and transcriptome responses to stress due to rain extreme events”